# Qualitative Analysis of Conversational Chatbots to Alleviate Loneliness in Older Adults as a Strategy for Emotional Health

**DOI:** 10.3390/healthcare12010062

**Published:** 2023-12-27

**Authors:** Antonia Rodríguez-Martínez, Teresa Amezcua-Aguilar, Javier Cortés-Moreno, Juan José Jiménez-Delgado

**Affiliations:** 1Department of Psychology, University of Jaén, Campus las Lagunillas s/n, 23071 Jaén, Spain; armartin@ujaen.es (A.R.-M.); cortes@ujaen.es (J.C.-M.); 2Department of Computer Science, University of Jaén, Campus las Lagunillas s/n, 23071 Jaén, Spain; juanjo@ujaen.es

**Keywords:** conversational chatbots, digital wellbeing, emotional health, loneliness, older adults, qualitative study, social support

## Abstract

This article presents an exploration of conversational chatbots designed to alleviate loneliness among older adults. In addition to technical evaluation, it delves into effective communication between these systems and this demographic group, considering linguistic nuances, communicative preferences, and specific emotional needs. The intrinsic importance of chatbots as innovative solutions in combating loneliness is highlighted, emphasizing their ability to be understanding and empathetic allies, contributing to emotional well-being and socialization. The article explores how improved emotional well-being can positively impact the health and quality of life of older adults. The methodology, rooted in triangulation between a literature review and qualitative research through interviews and focus groups with older adults, provides a comprehensive insight into the findings. Ethical, technical, and design considerations such as privacy, autonomy, technology adaptation, and usability are also addressed. The article concludes with practical recommendations for developing user-friendly interfaces that encourage the active participation of older adults in chatbots. This holistic approach not only analyzes the technical effectiveness of chatbots in mitigating loneliness in older adults but delves into human, ethical, and practical aspects, enriching the understanding and implementation of these agents for social and emotional support.

## 1. Introduction

In recent years, loneliness among the older adult population has become a growing global concern. Loneliness is defined as a perceived/subjective condition in which an individual is dissatisfied with the quality and/or quantity of their social relationships [1]. Loneliness can significantly impact the health and well-being of older individuals, increasing the risk of depression, anxiety, and chronic diseases [2]. Chatbots are an innovative tool to combat loneliness and enhance the quality of life for older adults [3]. Chatbots are artificial intelligence programs designed to interact conversationally with users, simulating natural conversation.

Chatbot technology plays a crucial role in mitigating loneliness in older adults by providing an interactive and personalized platform for communication and emotional support. The ability of chatbots to maintain meaningful and contextually relevant conversations offers older adults a virtual companionship experience that goes beyond mere mechanical interaction [4].

Chatbots can be designed to adapt to the individual preferences and needs of older adults, creating a conversational space that reflects each user’s personality and communication style. This personalization contributes to establishing an emotional connection, providing a sense of companionship that can be especially beneficial for those experiencing social isolation. Social isolation is an objective condition characterized by a lack of contact with other people and being disengaged from groups and social activities [1].

Furthermore, chatbots can play a crucial role in managing the mental and emotional health of older adults. By offering emotional support and positive reminders, these systems can counteract feelings of abandonment and contribute to psychological well-being. The ability to express thoughts and feelings openly, without fear of judgment, provides users with a safe space to share their concerns and joys, which can have therapeutic effects [5].

Everyday functions, such as medication and medical appointment reminders, represent another valuable application of chatbots in mitigating loneliness in older adults [6]. These systems can help ensure that users follow their medication regimens consistently, reducing the risk of unaddressed health issues due to forgetfulness or confusion. This practical aspect of chatbots contributes to improving the physical health of older individuals, strengthening the connection between emotional and physical well-being.

On the other hand, chatbot technology facilitates access to relevant and timely information, which can be especially valuable for older adults who may have difficulty accessing external resources. From news to health advice, chatbots can act as reliable sources of information, providing users with the opportunity to stay informed and connected to the world around them [7].

In the social sphere, chatbots can facilitate connections with friends and family, whether through messages or video calls. This ability to maintain and strengthen interpersonal relationships contributes to reducing the sense of isolation and fosters a sense of belonging and community [5].

Ultimately, chatbot technology emerges as a comprehensive tool for mitigating loneliness in older adults by combining emotional, health, and social aspects into an interactive and personalized experience. The ongoing evolution of this technology, guided by a user-centered approach and ethical considerations, promises to play an increasingly significant role in improving the quality of life for this vulnerable population.

The present study aims to comprehensively explore conversational chatbots designed to address loneliness in older adults, emphasizing their potential as empathetic allies for improved emotional well-being and socialization. It seeks to go beyond technical analysis, shedding light on ethical considerations and providing practical insights for the development of user-friendly chatbots. The ultimate goal is to offer a holistic perspective, enriching the understanding and application of chatbots as agents of social and emotional support for the older adult population.

## 2. Objectives

Throughout this article, we will explore the use and development of conversational chatbots as a solution to alleviate loneliness in older adults. The aim is to gather information that will help conversational chatbots become allies for older adults, somehow mitigating their loneliness. The primary purpose of this study is to undertake a detailed and enriching exploration of conversational chatbots specifically designed to address the issue of loneliness in the older adult population. Beyond a mere technical assessment, the goal is to comprehensively understand the crucial aspects related to effective communication between these systems and the demographic of older adults. This holistic approach aims to unravel the nuances of interaction and the specific emotional needs of older adults.

In this context, the article aims to highlight the intrinsic importance of conversational chatbots as an innovative solution in the battle against loneliness among older adults. It seeks to emphasize how these systems can transcend the technological realm to become understanding and empathetic allies that substantially contribute to the emotional well-being and socialization of this population. Delving into this aspect not only focuses on mitigating loneliness as an isolated problem but also aspires to explore how improving emotional well-being can have a positive impact on overall health and quality of life for older adults.

Furthermore, the article intends to shed light on the ethical and design considerations that prove imperative for the successful development of friendly and effective chatbots aimed at combating loneliness in this vulnerable population. Ethical considerations extend to privacy and autonomy, ensuring that the implementation of these systems is carried out with the utmost respect for individual rights and personal integrity. In the design realm, the importance of usability and accessibility will be addressed, intending to provide practical recommendations for the development of user-friendly interfaces that encourage the active and frictionless participation of older adults with chatbots.

This article seeks to be more than just a technical analysis of the effectiveness of chatbots in mitigating loneliness in older adults; it aims to delve into the human, ethical, and practical aspects of this innovative solution. Through this holistic approach, the goal is to enrich the understanding and implementation of chatbots as agents of social and emotional support in the lives of older adults, providing a comprehensive framework for future research and developments in this field.

## 3. Materials and Methods

The methodology of this study has been comprehensively designed, incorporating various research strategies to gain a deep and multifaceted understanding of the impact of chatbots on the older adult population and to gather information on the needs of this demographic in relation to the reduction or mitigation of their loneliness, both for individuals residing in their own homes and for individuals living in nursing homes. The methodological process is divided into three distinct phases, each contributing to the construction of a holistic framework that addresses both technical effectiveness and the individual needs and preferences of users.

This study commenced with a review of existing literature on the use of chatbots in the older adult population and loneliness. Subsequently, qualitative research was conducted through interviews and focus groups with older adults. Following this, an analysis was conducted to determine if there are studies evaluating chatbots to assess their effectiveness in reducing loneliness and enhancing emotional well-being in older adults.

The combination of these three phases provides a comprehensive view, addressing both the technical efficacy of chatbots and the needs and perceptions of the older adult population. Triangulating data from previous evidence [8], direct user feedback, and chatbot evaluation enhances the validity and applicability of the findings. This comprehensive methodological approach seeks not only to identify best practices in chatbot design for older adults but also to provide practical and ethical recommendations for the successful implementation of this technology in real-world settings. Each of the methodological aspects associated with the mentioned three phases is detailed below.

### 3.1. Review of Previous Approaches

The aim of this literature review was to analyze and synthesize existing evidence on the use of chatbots in the older adult population, focusing on identifying trends, previous approaches, perceived benefits, and associated challenges. The goal was to understand the effectiveness of chatbots in mitigating loneliness, improving emotional well-being, and promoting socialization among older adults. This literature review encompasses studies on chatbot evaluation to be considered in the final phase of the outlined methodology.

While systematic review steps have been followed to enhance rigor, the purpose is to gather supporting information through a narrative and thematic approach. Therefore, the approach used is that of a narrative review [9]. In any case, to conduct a narrative synthesis on trends, prior approaches, and identified gaps, not necessarily from all the works obtained in the search but from studies relevant to the broader context of this research.

Exhaustive searches were conducted in academic databases (related to social sciences, health, and engineering), specialized journals, and relevant conferences. Scopus, PubMed, and ProQuest databases were utilized. The search was conducted using terms and combinations such as “chatbots”, “conversational agents”, “older adults”, “elderly”, “loneliness”, “well-being”, and related terms. Filters were set to include only studies written in English or Spanish, with no temporal constraints. As an example, the search string used for the Scopus database is illustrated: “TITLE-ABS-KEY ((“chatbot” OR “bot” OR “conversational agent” OR “virtual assistant” OR “virtual agent” OR “digital assistant”) AND (“loneliness” OR “isolation” OR “aloneness”) AND (“older adult” OR “elderly” OR “senior”))”.

The included studies met the following eligibility criteria: (1) Focus on chatbots or conversational agents. (2) Involvement of older adults as the target population. (3) Relevance to the mitigation of loneliness, emotional well-being, or socialization. (4) Varied methodologies, including clinical trials, qualitative studies, quasi-experimental designs, and systematic reviews.

Two reviewers independently conducted this study selection. An initial review of titles and abstracts was performed to eliminate irrelevant studies. Subsequently, a comprehensive review of the full texts of the remaining studies was carried out to determine their final inclusion in the literature review.

Relevant data were extracted from each included study, including author, publication year, study design, sample size, chatbot intervention, main outcomes, and limitations. A standardized data collection form were employed. Data synthesis were conducted through a narrative and thematic approach, identifying emerging patterns in the results of the included studies and grouping information according to key themes. The compilation and synthesis of previous studies provided a solid knowledge base to inform this research design and contextualize the obtained results. 

### 3.2. Qualitative Research on the Needs of Older Adults

This study adopts a qualitative approach, necessitating a methodological stance of dialogical nature, wherein beliefs, mindsets, myths, prejudices, and emotions, among other elements, are embraced as analytical components to generate knowledge about human reality [10]. Building upon this, this research aims to delve into the experiences, perceptions, and social dynamics of older adults with diverse needs related to loneliness and its mitigation, as well as their acceptance of technology. By doing so, this study seeks to address the needs of these individuals in mitigating loneliness, ultimately enhancing their quality of life. The employed methodology involves both individual interviews and focus groups, enabling a thorough and holistic exploration of technological interaction and its emotional and social impacts. The focus are on describing life experiences, giving meaning to the analysis of collected data, and exploring the depth, richness, and complexity inherent to the addressed phenomenon. This approach allows for a profound exploration and the contribution of personal insights within the realms of trust and confidentiality.

Initially, this research considered conducting semi-structured interviews exclusively with older adults residing in nursing homes and those living alone in their own homes. However, challenges in accessing residents, coupled with the prolonged duration of interviews, particularly due to existing impairments, led us to contemplate the incorporation of two focus groups with older adults residing in their own homes to ensure a representative sample. Focus groups were not considered for nursing homes due to difficulties in accessing this study population.

These direct interactions allowed for a deeper understanding of the needs and preferences concerning loneliness and user experiences with technology. Individual interviews provided individual perspectives, while focus groups facilitated the exploration of group dynamics and the emergence of common themes. Special attention was given to the emotional and social perceptions of participants, as well as any perceived barriers to the adoption and acceptance of technology.

The population composing this sample has been deliberately selected to possess similar profiles and characteristics, ensuring the representation of diverse socioeconomic and cultural experiences. The goal is to enrich the understanding of the intersection between technology and loneliness in this demographic group, consisting of a total of *n* = 42 participants. All participants were considered within the age range of 65 to 80 years, with varying educational, formative, cultural, and professional backgrounds. Additionally, considerations were made regarding whether participants have descendants or close relatives. Both nursing homes and private homes, whether owned or rented, are located in central areas of cities with good transportation connections for accessing basic resources, parks, green areas, leisure zones, and walking spaces.

The sample studied in the interview process included a total of *n* = 26 older adults, with 20 residing in nursing homes and 6 living alone in their private homes.

For the selection of interviewees in nursing homes (*n* = 20), individuals with the ability to communicate, comprehend, and express themselves fluently were considered, excluding those with cognitive impairment or mental illness. This criterion aimed to ensure accessibility to the sample and guarantee the representativeness of their responses. Among them, 6 individuals (30% of the nursing home sample) are aged between 65 and 74 years, and 14 individuals (70%) are aged between 75 and 80 years. There are 12 women (60%) and 8 men (40%). The marital status includes 16 widowed individuals (80%), 1 married individual (5%), and 3 singles (15%). Regarding educational levels, 4 individuals (20%) in the sample are illiterate, 8 (40%) have basic literacy skills without formal education, 4 (20%) have completed primary education, 2 (10%) have completed high school or equivalent, and 2 (10%) have higher education and/or university degrees. Thirteen individuals (65%) have worked in sporadic and itinerant jobs related to agriculture, hospitality, and cleaning; 4 (20%) have worked in professions corresponding to their academic qualifications in positions of varying levels; and 3 (15%) have not worked in paid professions. All participants have children or close relatives who usually visit them.

The interviewed individuals residing in their private homes (*n* = 6) are in good physical and psychological health and possess the ability to communicate, comprehend, and express themselves fluently. Two individuals (33.3% of the sample) are aged between 65 and 74 years, and four individuals (66.7%) are aged between 75 and 80 years. Three individuals (50%) are men, and three (50%) are women. The marital status for all individuals is widowed. Housing arrangements included 3 individuals residing in rented apartments (50%) and 3 in owned apartments (50%). The educational levels of these interviewees include 2 (33.3%) with primary education, 2 (33.3%) with high school or equivalent, and 2 (33.3%) with higher education and/or university degrees. They have held jobs appropriate to their educational levels. Only one person (16.7%) does not have children or close relatives.

The interviews were conducted semi-structuredly, enabling participants to share their experiences openly and reflectively. Topics such as loneliness, influencing factors, how to cope with it, and whether technological support can meet the needs of this population, as well as potential technological barriers, were explored.

Following the individual interviews, this study extended its exploration through two distinct focus groups, each comprising eight participants, thereby forming a cohesive sample of 16 individuals (*n* = 16). The composition of these groups was carefully crafted, considering educational and cultural backgrounds to ensure a diverse and representative set of perspectives.

FG1: Older adults, residing independently. A total of 4 individuals (50% of the sample) were aged between 65 and 74 years, and 4 individuals (50%) were aged between 75 and 80 years. The gender distribution was evenly split between male and female participants. The marital status includes 5 widowed individuals (62.5%), 1 married individual (12.5%), and 2 singles (25%). Housing arrangements varied, with 2 individuals residing in rented apartments (25%), 3 in owned apartments (37.5%), and 3 in owned single-family houses (37.5%). Regarding education, 5 participants had higher education degrees (62.5%), and 3 had completed high school or equivalent (37.5%). The entire sample exhibited a medium-high cultural and economic level, having held professional roles in a medium-to-high category. In terms of family proximity, 7 participants did not have their relatives in the same locality or nearby (87.5%), and 1 individual had other relatives in close proximity (12.5%).

FG2: Similarly, this group comprised older adults, living independently, with an equal gender distribution. A total of 3 individuals (37.5% of the sample) were aged between 65 and 74 years, and 5 individuals (62.5%) were aged between 75 and 80 years. The marital status includes 6 widowed individuals (75%), and 2 married individuals (25%). Housing arrangements included 5 individuals residing in rented apartments (62.5%) and 3 in owned apartments (37.5%). Educational backgrounds varied, with 2 participants having no formal education but basic literacy (25%), 3 with primary education (37.5%), and 3 with secondary education (37.5%). The cultural and economic level leaned towards medium-low, and participants had itinerant work experiences linked to agriculture, cleaning, hospitality, and sporadic jobs. The entire sample had close family ties, with 7 participants having their relatives nearby (87.5%).

The aforementioned sample is summarized in Table 1.

Through the focus groups, a space for social interaction and collective discussion about experiences with technology was provided. This allowed for the identification of common patterns, contrasting perceptions, and the exploration of group dynamics.

Key themes explored included the key factors affecting loneliness, how to cope with it, the perception of technology’s empathy, associated emotions, and perceived barriers to its adoption. This flexible research design allowed for adaptation to unforeseen responses, enriching the understanding of the complexities of technological interaction in older adults.

Semi-structured interviews were conducted with the specified group of older individuals in the sample, using the script provided in Appendix A. The script in Appendix B was used for focus groups. The qualitative content of the information obtained from interviews and focus groups was analyzed using ATLAS.ti 23 software, where primary documents such as transcriptions, codes, annotations, and code groups were stored. The integration of findings from interviews and focus groups was undertaken, employing consistent categorization criteria. Despite the similarity in the lines of inquiry, the questions were thoughtfully adapted to suit the nuances of either interview or focus group settings. The synthesis of data resulted in the formation of distinct thematic groups, detailed in the results section of this study.

In the analysis, a thematic approach was adopted, starting with open coding to identify emerging patterns and categories. The collaboration of multiple researchers ensured a rich and contextualized interpretation of the collected data.

All participants provided informed consent, ensuring confidentiality and respect for their individual rights. Informed consent was prepared in writing, with assistance from the interviewer, who read and explained it to individuals with greater difficulty, continuously inquiring about their comprehension. In addition to obtaining verbal consent and ensuring clarity, the participant was then prompted to sign the consent form. This research was conducted with sensitivity and empathy toward participants’ experiences, ensuring that their voices were accurately reflected in the results.

### 3.3. Chatbot Evaluation

This study adopted a qualitative approach to assess the capability of chatbots, previously utilized in other studies, to provide companionship, emotional support, and practical assistance to older adults. This research focused on identifying the use of specific chatbots with the older adult population and exploring the effectiveness of the interaction and its emotional and social impact.

The selection of previous studies was based on a literature review, highlighting those that have explored the use of chatbots in interactions with older adults. The aim was to understand how these chatbots have addressed the specific needs of the population and whether they have provided comprehensive support.

Data analysis followed a qualitative approach, identifying emerging patterns and trends in the ability of chatbots to meet the specific needs of older adults. This methodology provided an in-depth understanding of how chatbots emotionally and socially impact older adults, enriching the existing literature on technological interaction in this population.

## 4. Results

In this section, we will elucidate the key findings derived from our comprehensive research approach. From the literature review to qualitative research and chatbot evaluation, each phase of our study has contributed to a profound understanding of the needs and experiences of older adults.

The review of previous approaches immersed us in the existing landscape of older adults’ interaction with chatbots. This exploration provided us with a solid foundation to contextualize and frame the results we present now.

At the core of our qualitative research are the direct voices and experiences of older adults. Through interviews and focus groups, we unraveled the complexities of loneliness, emotional and social perceptions, and potential barriers that may arise in the adoption of technology.

Furthermore, we assessed the effectiveness of chatbots used by older adults, focusing on their ability to provide companionship, emotional support, and practical assistance.

Across these phases, we aimed to shed light on the intricate intersection between loneliness, the emotional and social needs of older adults, and technology as a supportive tool. The results presented reflect the wealth of data collected throughout our multidimensional approach and provide a solid foundation for a better understanding of the role of chatbots in the well-being of this population.

### 4.1. Literature Review

Our literature review on the use of chatbots in the older adult population revealed several trends and previous approaches in the development of chatbots to address loneliness in older adults. Many studies have focused on designing conversational chatbots that provide companionship and emotional support to older adults. These chatbots were designed to simulate human conversations and offer a sense of connection and companionship to seniors experiencing loneliness. Additionally, some studies used chatbots to provide reminders of social activities and suggestions for participating in community events. These chatbots were intended to motivate older adults to engage in social activities and maintain an active and fulfilling life.

The literature search and study selection are depicted in the PRISMA diagram shown in Figure 1. In the review conducted, 134 studies were retrieved from the databases specified in the methodology. After removing duplicates, 82 documents entered the title and abstract screening phase. During this phase, 50 studies were excluded, leaving 32 studies for an in-depth review. Finally, 20 studies were included based on the eligibility criteria specified. These studies were analyzed in full text, extracting meaningful information regarding trends, design aspects, user perceptions, technology adoption, and user barriers. Additionally, information about specific chatbots tailored to older adults, used in the last phase of the methodology, was incorporated.

#### 4.1.1. Trends, Prior Approaches, and Gaps

In the current context, there is a growing demand for solutions aimed at supporting the older adults population in fundamental aspects such as health maintenance, social participation, and the promotion of independent living [12]. In this regard, virtual assistants emerge as an alternative to configuring intelligent and assisted environments that respond to the needs of older adults. An interesting aspect of this landscape is the use of sentiment analysis to evaluate the social acceptance of virtual assistants among older adults [12].

When reviewing prior approaches, it is highlighted that research has tended to focus on the technical aspects of virtual assistant development, addressing topics such as natural language processing and machine learning algorithms. Previous research has focused on the functional aspects of chatbots, such as their ability to provide reminders and manage medication [13].

Current trends include an increase in the use of chatbots for health coaching and social support, as well as an interest in personalizing interaction and content through natural language processing and machine learning techniques. In previous approaches, exploration has been made into the use of linguistic and emotional features, as well as avatars and robots as companions for older adults [14,15]. Nevertheless, we acknowledge that there has been limited exploration of the social and emotional aspects linked to the perception and acceptance of these assistants by older individuals.

Gasteiger et al. highlight the growing trend towards using technology-based interventions to address social isolation and loneliness in older adults [16]. This trend is reflected in the increasing number of studies on the topic and the development of new technologies designed to facilitate social interaction and communication.

In previous approaches to addressing loneliness in older adults, traditional interventions such as group therapy, social support programs, and volunteer programs have been employed. However, these approaches present limitations, such as restricted availability, high costs, and the need for transportation. Social robots and computer agents offer a potential solution to these limitations by providing a more accessible and cost-effective intervention that can be used in the home environment [17].

Although research has revealed important factors in the acceptance of social robots, older adults, especially those aged 85 and older, have been underrepresented in most studies [18]. This underscores a gap in research in not fully understanding the attitudes and experiences of the older population towards social robots. Furthermore, it is noted that previous research has focused on the effects of robot communication strategies in advice-giving situations, but there is a need to delve into the use of social robots as companions for older adults. The study by Fakhr Hosseini et al. highlights a research gap in not fully understanding the potential benefits and limitations of social robots as companions for older adults. In summary, they suggest that more research is needed to better understand the attitudes and experiences of older adults towards social robots and to develop effective strategies for designing social robots that meet the unique needs and limitations of this demographic group [18].

Some authors emphasize crucial aspects, such as the increasing personalization of chatbots for interaction with older adults and adapting language and functions to the user’s personality and needs. They identify the need to investigate the cognitive impact of chatbots on older adults, highlighting the shortage of solutions focused on cognitive exercises [19].

Despite the potential benefits of social robots and computer agents, several research gaps are highlighted in the field [16]. One such gap is the lack of standardization in the design and implementation of these interventions, making it difficult to compare results across studies. Another gap is the limited research on computer agents, as most included studies focused on social robots. It also emphasizes the need for more research on the long-term effects of these interventions as well as their effectiveness in different populations of older adults. Additionally, it is suggested that future research should explore the potential negative consequences of these interventions, such as increased social isolation or dependence on technology [16].

The lack of standardization in the evaluation of chatbots requires holistic analyses of their impact on the lives of older adults [16]. Additionally, the need to investigate the accessibility of chatbots for those with physical and cognitive limitations is emphasized by Pinto et al., seeking solutions that encompass diverse abilities [19]. Finally, the importance of delving into research on the emotional impact of chatbots on older adults is highlighted, aiming to design solutions that provide meaningful and effective support in this aspect [19].

In the landscape of future research, there is a need to delve into the social and emotional aspects related to virtual assistants, with a focus on how to design these technologies to adapt to the individual needs and preferences of older users [12,20]. Likewise, the importance of further research on the use of sentiment analysis to assess social acceptance is emphasized [12], considering how this tool can be used to improve both the design and functionality of virtual assistants. Additionally, there is an urgency to explore the ethical implications associated with the use of virtual assistants in the care of older individuals, focusing on issues such as privacy, security, and autonomy [12]. The need for broader studies to better understand user behavior in chatbot environments, investigate the impact on social relationships, and explore ethical implications for older adults is emphasized [14].

A summary of these trends, approaches, and gaps can be seen in Table 2.

#### 4.1.2. Design Aspects of Chatbots

Certain studies focus on social robots as companions for older adults, providing valuable insights into the functions that older adults attribute to a companion robot, such as expressing emotions, understanding emotions, initiating conversations, and walking with them. These findings connect with the idea that chatbots designed for older adults could benefit from incorporating features that provide emotional support and encourage social interaction. In the study by FakhrHosseini et al., it is emphasized that older adults expressed frustration due to the limited ability of the robot to engage in real conversations and the restrictions on social communications [18]. These aspects reinforce the importance of developing chatbots that not only fulfill practical functions but also address the emotional and social needs of older users [15].

In this context, it is noted that chatbots should be designed to be engaging, supportive, and emotionally intelligent, using conversational prompts and feedback tailored to the needs and preferences of the user [14]. Concrete examples are proposed, such as incorporating measures of accessibility and usability, such as larger fonts, clear language, and intuitive navigation, considering the possible cognitive or physical limitations of older adults. Additionally, the importance of making chatbots easy to use and understand for older adults with varying levels of technological literacy is highlighted. This study also suggests that chatbots should be engaging, supportive, and emotionally intelligent, providing emotional support and validation when needed, and using humor and social cues appropriately to build rapport and connection.

Pinto et al. emphasize the fundamental importance of personalizing the interaction of chatbots with older adults [19]. To achieve this, it is proposed that these systems adapt to the personality, interests, and cultural context of the user, also considering their physical and cognitive limitations. This approach involves providing personalized recommendations and reminders, as well as adjusting the language and style of the chatbot according to individual preferences. Furthermore, it advocates for designing chatbots that are intuitive and accessible for older adults. This includes using simple and clear interfaces, concise instructions, and leveraging technology to improve accessibility, especially for those with physical or cognitive limitations [22]. The relevance of emotional support in interacting with older adults through chatbots is also emphasized. The ability of these systems to offer companionship and emotional support is highlighted, especially valuable for those experiencing social isolation or loneliness.

Another crucial aspect addressed in the literature review is the importance of personalizing the interaction of chatbots with older adults. Adapting to the personality, interests, and cultural context of users, as well as their physical and cognitive limitations, is presented as a key strategy for creating more engaging and meaningful experiences [19]. The importance of feedback and user testing in the development of chatbots for older adults is emphasized. The active involvement of this demographic group in the design and testing process ensures that chatbots effectively meet their needs and preferences. This comprehensive approach is proposed as key to the successful development of chatbots that truly enhance the quality of life and emotional well-being of older adults [19].

Design aspects are summarized in Table 3.

#### 4.1.3. Emotional and Social Perceptions of Older Adults

In previous studies, key results based on the attitudes and experiences of older adults towards social robots were highlighted [18]. It is noted that the acceptance of these robots as companions was positively influenced by exposure and limited interaction with the robots. Regarding emotional and social perceptions, some older adults attributed functions to the companion robot, such as expressing emotions, understanding them, and initiating conversations.

User behavior, such as expressiveness, feelings, and personal disclosure, is analyzed in the study conducted by Razavi et al. It describes how such behavior is linked to the topic under discussion and the tone of the avatar, observing its evolution over time. This study provides valuable information about the needs, preferences, and experiences of older adults in relation to chatbots. This research suggests that chatbots intended for older adults should adapt to the needs and preferences of users, offering appropriate feedback and support [14].

The study by Troncone et al. (2020) emphasizes the importance of considering the emotional and social perceptions of older people in the design and implementation of support technologies [23]. The authors suggest that older people’s attitudes towards technology are influenced by various factors, such as perceived usefulness, ease of use, and social influence. The importance of designing support technologies that are friendly, personalized, and socially acceptable to enhance the engagement and satisfaction of older individuals with these technologies is also discussed.

Other studies delve into social perceptions, finding that older adults value the ability to share health information with family and healthcare professionals [24]. This study suggests that they perceive technology as a facilitator of communication and collaboration with others.

In a study conducted with the Care Coach, users’ perceptions of the relationship with chat agents were highlighted [16]. It is mentioned that some participants considered the relationship to be superficial due to the limited ability of the agent to engage in conversations. However, users reported that the presence of the robot or agent gave them the feeling of having “someone” there for them and “someone” to talk to, thereby reducing the feeling of loneliness. This study suggests that users may value the ability of chatbots to offer companionship and conversation. However, the need for further research to better understand the specific needs and preferences of users in relation to chatbots is acknowledged.

On the other hand, it has been observed that older users may experience anxiety and frustration when using chatbots, especially if they are unfamiliar with technology. Additionally, this study highlights that older users may hesitate to adopt new technologies due to concerns about privacy and security [13].

#### 4.1.4. Technology Adoption and Utility

Regarding technology adoption, previous studies focused on social robots. The acceptance of companion functions was more evident in the older adult group than in other demographic groups, although there was no correlation between previous technological experience and the acceptance of social robots [18]. The previous study suggests that emotional and social perceptions of robots as companions were generally positive, regardless of the level of technological experience [18].

Other studies revealed that older adults showed a general receptiveness to the use of intelligent assistants for health information management, expressing the desire for these to be personalized according to their individual needs and preferences [24]. This study suggests a positive emotional response to the idea of using technology to improve their health outcomes. However, concerns about the accuracy and reliability of information provided by intelligent assistants were also identified, indicating some anxiety or uncertainty about using technology for health information management.

Pinto et al. highlight various ways in which chatbots can play a significant role in the lives of older individuals [19]. First, their ability to provide emotional support and companionship is emphasized, which is especially valuable for those facing social isolation. Chatbots can offer a listening ear, words of encouragement, and a friendly presence. An essential component addressed is the role of chatbots in fostering social interaction and engagement. By providing opportunities for conversation and socialization, chatbots can contribute to counteracting isolation and loneliness, common issues in the lives of older individuals [19]. In the realm of cognitive health, it is suggested that chatbots can play a crucial role in providing cognitive exercises and educational content tailored to the user’s interests.

#### 4.1.5. Barriers to Technology Use

Concerning barriers to technological adoption, concerns about cost, potential technical difficulties, privacy, and security were identified. In the study by FakhrHosseini et al., concerns about affordability, the possibility of technical issues, and the need to ensure privacy and security are revealed [18]. These findings emphasize that cost, technical difficulties, and concerns about privacy and security could be considered obstacles to the adoption of social robots among older adults.

Another study highlights perceived barriers to technology adoption and acceptance, such as concerns about privacy and security, as well as a lack of familiarity with technology [24]. Participants in this study expressed a desire for smart assistants to be easy to use and understand, indicating a perception of technology as complex or difficult to navigate.

Troncone et al. identified several perceived barriers to the adoption and acceptance of technology by older adults, including the stigma associated with technology, threats to autonomy and privacy, and a lack of adequate information and training [23]. The authors suggest that addressing these concerns and providing appropriate information and training can increase the acceptance of technology by older individuals.

On the other hand, it is suggested that chatbots can overcome barriers to technological adoption among older individuals. By providing intuitive interfaces, clear instructions, and personalized recommendations, these systems can make technology more accessible and user-friendly for this demographic [19].

Below is a summary (Table 4) of emotional and social perceptions, technology adoption, and its utility, as well as barriers found in the literature.

### 4.2. Qualitative Research on the Needs of Older Adults

In this section, the results derived from a detailed analysis of individual interviews and focus groups conducted with older adults are presented. This study focused on exploring key themes, including determinants of loneliness, coping strategies, perceptions of technology empathy, associated emotions, and perceived barriers to technology adoption in this demographic group. Through a robust qualitative methodology, the aim was to gain a deep and holistic understanding of technological interaction and its emotional and social impacts on the experience of loneliness in older adults.

The detailed analysis of interviews and focus groups reveals a rich diversity of needs among older adults in their experience of loneliness. Fundamental themes emerge, shedding light on the complexities and nuances inherent in these needs. 

Among the interviewed individuals, we highlight the following expressions:

E2: *“[…] what I need is someone to talk to […], when I find myself so alone at night in my house, I think, what if something happens to me and I can’t even press the button, who will find me here, and above all when […]”*

E4: *“[…] experience tells me that it already happened to my mother and father, my children are not nearby, I only have a few acquaintances in the neighborhood, who are becoming fewer, and every day I’m more alone, when one finishes their work stage and, on top of that, your wife goes to heaven […] everything had to end for me too. My life today is very sad.”*

On the other hand, among the focus groups, we could highlight a less profound reference:

FG2.1: *“[…] my day-to-day is not very exciting, but I’m getting by, I have moments of everything, but today I’m happy because we are talking here […]”*

These results provide a holistic insight into the complex needs of older adults in relation to loneliness, offering valuable insights for the design of comprehensive interventions that effectively address this reality.

#### 4.2.1. Company and Social Connection

The importance of companionship and social connection is highlighted as an essential element to mitigate loneliness, according to Baker et al. (2018) [25]. Participants emphasize the need for meaningful interactions, whether with family, friends, or new relationships, as a means to cultivate a sense of belonging and reduce the feeling of isolation. 

E20: *“[…], it’s just that my life is different now. Even though I’m surrounded by people here, it’s not my home, next to my neighbor, where I raised my children and where I have my acquaintances. I used to go out to the square, and just talking to others there gave me life. But of course, my children are far away now, and my husband is no longer here, and that’s why I’m here. I’m fine here, but the truth is that I feel more alone than in my own home […]”*

These interactions are not limited to the physical presence of others but encompass the quality of emotional connection and meaningful communication. For many, the company of close family and friends plays a crucial role, as stated by Dosovitsky & Bunge [26]. Engaging in shared activities, such as meals, walks, or simple conversations, represents moments when they feel connected and supported. The ability to share experiences and express thoughts and emotions creates a social fabric that mitigates the feeling of isolation.

E15: *“[…] I was feeling so-so at home because most days I didn’t talk to anyone. I need to go out, talk, meet with relatives, with work friends. The truth is, my children don’t leave me much, but at least they call me on the phone, if it’s not one, it’s the other, and my granddaughters, they make video calls, and at least I see them. But, in the end, I was alone at home […], and I told my daughters that I was coming to the residence because it’s in the same neighborhood, and I can almost stick to my routines, but now I’m taken care of. If there’s an emergency here, it’s better, plus, we have many activities, and I do things that cheer me up.”*

Additionally, some participants mentioned the importance of cultivating new relationships and expanding their social circles. This responds to the reality that, in certain cases, existing social networks may decrease due to factors such as reduced mobility, the loss of friends or family members, or changes in life circumstances.

Social activities, community clubs, or local events were also mentioned as valuable ways to foster social connection. These activities not only provide a structure for interaction but also offer opportunities to meet new people and build relationships, thus contributing to a sense of belonging and connection.

FG1.6: *“[…] the thing about having my activities every day, sports, the association, those things that, even though they are not work, I do now because I want to, and they keep me active and in touch with the world.”*

Company and social connection not only encompass the physical presence of others but also the quality and depth of interpersonal relationships. It provides a solid framework to counteract loneliness, creating an environment where older adults feel emotionally nurtured, supported, and part of an active social network.

#### 4.2.2. Emotional Support

The analysis revealed that emotional support stands out as a critical need for older adults in their struggle against loneliness, as previously indicated by Dosovitsky & Bunge [26]. Participants express the longing to have someone to share their emotions with, establishing the presence of a supportive figure who can understand and validate their feelings as fundamental to coping with loneliness, according to Biallas et al. [27]. There exists an undeniable desire for emotional support, both in challenging times and in their day-to-day lives.

E2: *“[…] I need to go out every day, I go to the club, I have my friends, we meet for breakfast in the center, then we stroll through the shops, something usually catches our eye, some days we even meet for lunch, well […]. My reality hits at night because we only meet on weekends and some holidays, and of course, the rest of the days, the nights become long.”*

In our findings, family, friends, and loved ones played a prominent role in providing emotional support. The ability to share their thoughts, concerns, and joys with trusted individuals is perceived as an effective balm against loneliness. Empathy, understanding, and the simple act of listening were identified as the fundamental elements of this emotional support. However, some participants pointed out that, at certain times, the availability of friends and family may be limited due to various circumstances, such as geographic distances or personal commitments. It is in these contexts that the need for emotional support takes on a particular nuance.

Technology emerged as a potential resource to provide emotional support, according to other authors [27]. In our study, participants expressed interest in technological solutions that could offer sensitive and empathetic companionship. The idea of having a virtual support system that can provide words of encouragement, understanding, and, in some cases, even trigger playful or reflective interactions was well-received.

FG1.4: *“[…] look, what really gets to me is when ALEXA says, ‘you are a great human,’ […] how can this device talk to you like a human, great, […], well, it’s a device and it says things without feelings, with an understanding of what you tell it or express to it at that moment.”*

E3: *“[…] I started by just using these things from social networks, and then I took a course where they really taught me how to use them. That’s where I learned that what really limited me was the difficulty of the unknown. Now, every day, I meet more people, and above all, I have my loved ones closer.”*

Emotional support, from this perspective, is not only conceived as a response to difficult situations but also as a constant presence that enriches everyday life. The ability to feel understood, valued, and emotionally supported is perceived as an essential component of counteracting loneliness and promoting the emotional well-being of older adults.

#### 4.2.3. Meaningful Occupation

The notion of meaningful occupation emerged as a relevant theme, highlighting the importance of meaningful and rewarding activities to combat loneliness in older adults. Participants emphasized the need to occupy their time meaningfully, feeling productive and connected to their interests and passions.

FG1.5: *“[…] well, let me tell you, in this, I go further. My life has been very active, and what I need is not only to avoid feeling alone in many moments, but also to have resources that allow me to discover new things, so that my time is satisfied and rewarded every day by doing new things. Today, the technological aspect has advanced a lot, and it is important to create things for older people as well, as not everyone is like me.”*

Engaging in meaningful activities was considered essential to maintaining a sense of purpose and contributing to emotional well-being. Occupations that offer a challenge, stimulate creativity, and generate a sense of accomplishment were identified as particularly beneficial. These can range from artistic and craft activities to participation in community groups or teaching skills to others.

FG1.6: *“[…] it helped me a lot to see other people who record their videos and share many activities and skills that the rest of us can do.”*

However, it was emphasized that, in some cases, loneliness can hinder access to opportunities to engage in meaningful activities. Factors such as reduced mobility, a lack of local opportunities, and economic limitations were mentioned as barriers to participation in occupations that could be meaningful.

FG2.5: *“[…] when I fell and broke my hip, that’s when I was truly alone; I couldn’t do anything interesting or talk to anyone […]”*

Technology was considered a potential means to facilitate meaningful occupations. The idea of accessing online platforms that offer learning opportunities, participation in virtual communities, or engaging in creative activities was met with interest. The possibility of connecting with others who share similar interests and participating in meaningful activities through technology was perceived as a valuable tool to mitigate loneliness.

#### 4.2.4. Access to Health and Well-Being Resources

Accessibility to health and well-being resources was identified as a critical dimension in combating loneliness among older adults, as can be seen in the literature [28]. The importance of having easy access to healthcare services, wellness programs, and health-related information was emphasized through our study.

FG1.8: *“[…] when I go out to buy groceries, well, actually, I go out to get bread, but I say groceries; I do it to see people and talk to someone, it’s my little chat time. […] but what gives me the most trouble is scheduling appointments with my doctor, that’s why I go to the health center […]”*

Mobility and proximity to healthcare services were considered key variables for effective access to health resources, as stated by Chou et al. [28]. Participants expressed that the ability to easily attend medical appointments, wellness centers, and group activities was fundamental to maintaining good physical and mental health. Lack of mobility, whether due to physical limitations or geographic restrictions, was identified as a significant challenge. Those facing difficulties in movement reported feeling more isolated and experiencing greater barriers to accessing health and well-being services. Dependence on others for transportation was also noted as a limiting factor.

E16: *“[…], not anymore because I’m here, but before, in my house, every time I had to go to the doctor, it was two or three trips, and for the bank, I took the approach of going in the middle of the month because it was impossible.”*

Technology emerged as a potential facilitator to overcome some of these barriers. The possibility of accessing health services online, participating in virtual consultations, and receiving relevant well-being information through digital platforms was considered a valuable solution for those with mobility limitations.

FG1.4: *“[…] I started using video calls and some applications for interactive games a while ago, but when COVID hit, I learned even more, and today it makes it easier for me to schedule a phone appointment and have a consultation through a video call […]”*

The importance of having access to up-to-date information on health and well-being topics was also highlighted. Participants expressed a desire for resources that provided practical knowledge on healthy aging, management of medical conditions, and strategies to improve their overall well-being.

FG2.6: *“[…] I don’t understand much about this, but it seems important that these things are also taught to older people; we are also in the world.”*

#### 4.2.5. Safe Environment and Personal Care

Creating a secure environment and providing personal care are presented as essential requirements to cope with loneliness. Feeling secure in their surroundings and having access to care that supports their physical and emotional well-being are revealed as crucial aspects based on shared experiences.

E17: *“[…] I am here because for me and for my children, the important thing is that I am taken care of when I need it. I had a stroke, and I was alone until my daughter arrived and could come to me. But that was lucky; I need to be taken care of, and that’s why I’m here […]. Many times, we think our children are close, but things happen, and we are alone. That’s why I thought what I thought and said, I’d better be in a nursing home.”*

The results revealed the importance of a safe environment and the role of personal care in the lives of older adults, especially in the context of addressing loneliness.

Safety in the environment was highlighted as a critical need. Participants expressed the importance of living in places where they felt protected and comfortable. The presence of security systems, suitable housing conditions, and friendly environments were identified as fundamental elements to promote emotional well-being and alleviate feelings of loneliness.

E1: *“[…] for me, it is very important to be in company, but more importantly, it is to feel secure in my home, knowing that if something happens, someone will come to me.”*

Personal care emerged as a significant theme in the analysis. Participants described how support in daily activities, healthcare, and the companionship of caregivers or family played an essential role in their emotional well-being and perception of loneliness. Those who had access to a personalized care system reported feeling more supported and less lonely.

E13: *“[…] you see, if I had had a companion, a device that reminds me of my medication, reminds me of my medical appointments, the birthdays of my daughters and grandchildren, […] I would stay in my home for more years.”*

Technology was also mentioned in relation to safety and personal care. Participants valued technological solutions that allowed them to access help quickly in emergency situations, as well as monitoring devices that provided peace of mind for both them and their loved ones.

FG1.7: *“[…] as I’ve been saying, I have been using many technological resources for a long time, but what I mean is, with so many things available today, why aren’t more effective things done for us? So that we feel cared for, attended to, with control of appointments and medication, […], or things like that. I know this can be done, but I haven’t seen it, and it would be a very good thing for people like us.”*

The need to maintain autonomy in personal care was a recurring theme. Participants expressed the desire to be able to carry out everyday activities independently, and some mentioned technology as a means to facilitate that autonomy, such as automatic medication reminders or assistive devices.

#### 4.2.6. Technological Adaptability

Technological adaptability emerges as a pressing need. Although some participants are open to technology as a means to connect with others, there is a clear demand for intuitive, accessible, and tailored technological solutions that align with their abilities and preferences, as indicated by Jegundo et al. [20]. This issue emerged as a fundamental theme during the analysis, revealing how older adults perceive and experience technology in their daily lives. This dimension aligns with Baker et al., emphasizing the willingness and ability to adopt and use technological tools, particularly in the context of addressing loneliness [25].

FG2.4: *“[…] since COVID, I started using my tablet to access some of these online gaming sites and social networks. Honestly, I feel more accompanied, and I meet people […]. It’s true that you don’t know these people in person, but at least you talk to someone. And then, I combine it with some outings, friends, and family. Especially my daughter, who tells me to be careful with social networks […].”*

The results highlighted a variety of attitudes toward technology among participants. Some expressed a positive openness and genuine interest in learning and using technological devices, while others showed some resistance or anxiety related to the use of new technologies.

FG2.4: *“[…] for me, it’s very difficult to use these things, but of course, I understand that they can help us. But I think it’s very difficult for me to learn it now.”*

Accessibility and usability were key factors influencing technology perception. Participants emphasized the importance of simple interfaces, intuitive features, and user-friendly designs to facilitate technological adaptation. Those who experienced difficulties with complex or unintuitive devices expressed greater reluctance toward technological adoption.

FG1.4: *“[…], ideally, we should have easy-to-use resources that we can use.”*

Training and support in using technology emerged as crucial elements to improve adaptability. Participants who had received guidance or training on the use of technological devices showed greater confidence and willingness to incorporate these tools into their daily lives.

FG1.6: *“[…], look, I have been using computers for a long time, because of my job, but I tell you that there are many things I don’t know how to do. Many new things come up that I don’t use because I don’t know how they work. If they explain things to me well, I could definitely do them […].”*

The perception of the potential benefits of technology also influenced attitudes toward technological adaptation. Those who recognized advantages, such as the ability to stay connected with friends and family, access relevant information, and participate in online activities, were more likely to overcome barriers and adopt technologies.

In Table 5, a summary of the various identified needs of older adults is presented.

### 4.3. Chatbot Evaluation

In this section, we will proceed to conduct a detailed evaluation of various conversational chatbots specifically designed to address the needs of older adults. The evaluation will focus on analyzing the effectiveness of these technological tools in providing companionship, emotional support, and practical assistance to this demographic.

The selected chatbots represent a variety of technological approaches, interaction styles, and levels of complexity and have been identified based on the literature review performed by CATe [29] and Charlie [21], as well as other sources of information from ElliQ (https://elliq.com/, accessed on 20 November 2023).

#### 4.3.1. CATe Chatbot

CATe is an Embodied Conversational Agent designed to provide companionship and emotional support to older adults [30]. The evaluation form used by Bravo et al. to assess CATe includes categories such as naturalness, embodiment, interaction and affect, enjoyment of use, ease of use, acceptability, and utility [29]. These categories are used to assess how human-like the virtual agent is when interacting with older adults, how well the agent can convey sympathy through facial expression, speech, and gestures, and how well older adults accept and adopt the technology. The evaluation results from Bravo et al. show that older adults generally enjoyed talking to CATe and liked the application, achieving an overall satisfaction score of 3.98 out of 5 [29]. The aforementioned study also alludes to the fact that some of the older participants had difficulty understanding English, and some evaluation conversations and questions had to be translated into the local language by the researchers. Additionally, it mentions that some participants needed assistance due to their technological limitations.

#### 4.3.2. Charlie Chatbot

Another study was designed to investigate interaction design strategies to develop a chatbot named Charlie that can engage in dialogue with older adults and help improve the quality of their lives in psychosocial terms [21]. This study aimed to investigate the attitude, acceptance, enjoyment, and utility of using a specific set of functionalities mediated by a chatbot to encourage the desire to feel good and do good. The authors found that participants had positive emotional and social perceptions of the chatbot. They reported feeling happy, entertained, and less lonely when interacting with Charlie. Participants also perceived Charlie as a friendly and helpful companion who could provide useful information and support. However, the authors also identified some perceived barriers to the adoption and acceptance of technology. For example, some participants reported difficulties understanding the instructions and functionalities of the chatbot. Others expressed concerns about the privacy and security of their personal information. The authors also acknowledged that the participant sample size was small and that this study was conducted in a specific context, which could limit the generalization of the findings [21].

#### 4.3.3. ElliQ Chatbot

ElliQ (https://elliq.com/, accessed on 21 November 2023) has been designed to consider the specific needs of older adults. Its design incorporates elements that facilitate interaction, such as facial expressions, movements, and lights. These aspects are crucial to making the interaction intuitive and comfortable for a demographic that may not be familiar with more complex technologies. ElliQs adaptability is also a key component of its design, as it can learn and adjust to the individual preferences and routines of users.

ElliQs ability to express itself through movements and facial expressions contributes to users’ emotional perception. By answering questions, telling jokes, and engaging in conversations, ElliQ seeks to establish an emotional connection and provide companionship. Natural interaction and a focus on communication contribute to the perception of ElliQ as a social companion, which can be especially valuable for mitigating loneliness in older adults.

ElliQ offers practical utilities, such as reminders and alerts, that are essential for the daily needs of older adults. The ability to provide useful information, entertainment, and facilitate communication with family members extends its utility. Furthermore, its ability to adapt to individual preferences enhances its practical value, as it can customize its interaction based on the specific needs of each user.

The adaptability and socially focused features of ElliQ position it as an effective tool for mitigating loneliness in older adults. By providing companionship through conversations, entertainment, and facilitating communication with family members, ElliQ addresses emotional and social aspects related to loneliness. The ability to learn and adjust over time also suggests a tool that can evolve to meet the changing needs of users as they age.

## 5. Discussion

In our comprehensive research, fundamental findings have emerged from a multifaceted approach encompassing a literature review, qualitative research, and chatbot evaluation. Each phase has contributed to a deep understanding of the needs and experiences of older adults, providing a comprehensive insight into the intersection between loneliness, emotional and social needs, and technology as a support tool.

The literature review immersed us in the existing landscape of older adults’ interaction with chatbots, providing a solid foundation to contextualize our findings. Trends in chatbot development to address loneliness were identified, highlighting increasing personalization and interest in emotional and social aspects, as noted by several studies [18,23]. However, research gaps were also noted, such as the lack of standardization in intervention design and the need to explore the long-term consequences of these technologies, in line with Troncone et al. [23].

In the qualitative phase, we delved into the voices and direct experiences of older adults through interviews and focus groups. This allowed us to unravel the complexities of loneliness, emotional and social perceptions, as well as barriers that might arise in technology adoption. Participants emphasized the importance of companionship and social connection, emotional support, meaningful occupations, access to health and wellness resources, a safe environment, and personal care.

The growing demand for solutions to support the older population has driven the development of virtual assistants as alternatives in smart environments. Previous research has prioritized technical aspects, such as natural language processing and machine learning, with a recent focus on health coaching and social support chatbots [31]. Despite exploring linguistic and emotional features, as well as the use of avatars and robots as companions, research on the social and emotional aspects related to perception and acceptance by older adults has been limited [12,20]. The trend towards technological interventions to address social isolation highlights the utility of social robots and home agents, but the underrepresentation of older adults, especially those aged 85 and above, reveals a gap in understanding their attitudes [18]. The lack of standardization in chatbot design, implementation, and evaluation emphasizes the need for future research focused on social and emotional aspects to inform more effective and user-centered design, as discussed by FakhrHosseini et al. [18].

According to our findings, effective chatbot design for older adults focuses on personalization, adapting to personality, interests, and cultural contexts, as well as physical and cognitive limitations. It is crucial for chatbots to be engaging, supportive, and emotionally intelligent, using tailored conversational prompts. Accessibility and usability measures, such as large fonts and clear language, are proposed. Personalization, including tailored recommendations and reminders, is presented as essential. Feedback and testing with active participation from older adults are key to successful design, ensuring that chatbots effectively meet their needs and preferences, as indicated by several studies [19,22].

Perceptions of older adults about chatbots are positive when they experience companionship and conversation, reducing the sense of loneliness. However, a lack of familiarity with technology can lead to anxiety. Concerns about privacy and security are also potential barriers. The adaptability of the chatbot to tone and topic, along with its ability to provide emotional support, is highlighted as essential for its success as a companion for older adults. Previous findings align with those of other authors in the literature [18,23,24].

Older adults in our study show overall receptiveness to intelligent assistants for health information management, emphasizing the importance of personalization. Acceptance of the company of social robots correlates with previous technological experience, but emotional and social perceptions are positive overall. The perceived utility of chatbots stands out in providing emotional support, companionship, and adapted cognitive activities, countering social isolation, and promoting cognitive health, as indicated by Biallas et al. [27].

Older adults face obstacles such as affordability, technical issues, and privacy concerns when adopting technology. Lack of familiarity and anxiety about information accuracy are key barriers [24]. Overcoming technological stigma, addressing threats to autonomy, and providing appropriate information and training are proposed strategies to increase acceptance. Martin-Hammond et al. suggested that chatbots can be helpful by offering intuitive interfaces and personalized recommendations, making technology more accessible, and enhancing the emotional and social well-being of older adults [24].

The qualitative study on the needs of older adults provides a comprehensive insight into the factors influencing loneliness in this demographic group. Through individual interviews and focus groups, various dimensions were explored, highlighting the importance of companionship and social connection. Participants emphasized the need for meaningful interactions, whether with family, friends, or the possibility of establishing new relationships. The quality and depth of these interactions were stressed as essential to counteracting loneliness, indicating the importance of addressing not only the physical presence but also the emotional relevance of social relationships. Additionally, this study identified the critical relevance of emotional support and meaningful occupation in the fight against loneliness, according to Dosovitsky & Bunge [26]. Participants recognized the limitations of the availability of friends and family at certain times, highlighting technology as a possible solution to provide constant emotional support. Meaningful occupation emerged as an essential need to combat loneliness, and technology was acknowledged as a valuable tool by offering online opportunities for learning, participation in virtual communities, and engaging in creative activities, as mentioned by Biallas et al. [27].

This qualitative study addressed the importance of various aspects, such as access to health and wellness resources, a safe environment, and personal care. Technological adaptability emerged as a fundamental theme, highlighting variability in participants’ willingness and ability to adopt and use technological tools. This finding underscores the importance of designing intuitive technological solutions and providing the necessary support to encourage their adoption. Collectively, these results inform the need for comprehensive interventions that address not only the social and emotional dimensions but also technological adaptability and access to fundamental resources to improve the quality of life for older adults, aligning with the recommendations made by Baker et al. [25].

The evaluation of chatbots reveals the diversity of technological approaches and their effectiveness in addressing the needs of older adults. CATe, an Embodied Conversational Agent, showed an overall positive response, but language difficulties and technological limitations were highlighted [29]. Charlie, designed to enhance the psychosocial quality of life, generated positive perceptions, although barriers such as comprehension difficulties and privacy concerns emerged [21]. ElliQ (https://elliq.com/, accessed on 21 November 2023), specifically designed for older adults, stood out for its focus on natural interaction and adaptability, addressing both emotional and practical aspects. The evaluation performed in our study underscores the importance of considering the diversity of user skills and preferences, as well as carefully addressing privacy and security concerns in the design of these technologies. Overall, our results indicate significant advancements in chatbot design for older adults, with opportunities for further improving accessibility and usability.

### 5.1. The Ideal Chatbot Based on Conducted Studies

Drawing from the insights garnered in this study, we can distill essential considerations for crafting a conversational chatbot tailored to the unique needs of older adults, with the primary aim of mitigating their experience of loneliness. An ideally designed chatbot, as envisioned here, goes beyond mere functionality, aspiring to be a thoughtfully engineered virtual companion that addresses not only loneliness but also other critical dimensions of life. The ensuing features are elucidated based on a comprehensive synthesis of findings derived from our study, encompassing insights from the literature review, qualitative research, and chatbot evaluation. The following features are presented based on our study’s findings. 

Personalization and Adaptability, as highlighted in various studies [18,23,24]. Tailoring interaction to the user’s personality, interests, and cultural context is essential for creating meaningful experiences. This involves adjusting conversation tone, speed, and level of detail according to individual preferences. Adaptability also refers to the chatbot’s ability to accommodate users’ physical and cognitive limitations, ensuring an inclusive experience.

Empathy and Emotional Sensitivity are in line with Lappromrattana & Sooraksa [15]. The chatbot’s ability to express empathy and emotional sensitivity is crucial for providing constant emotional support. This includes incorporating responses and feedback that reflect understanding and affection, thus contributing to reducing the feeling of loneliness. Attention to emotional needs, such as anxiety and frustration, should also be present in the design of responses and interactions.

Cognitive Stimulation and Meaningful Occupation: Several studies, have outlined this [26,27]. The chatbot should offer activities that stimulate the mind, promoting meaningful occupations. This may include providing cognitive exercises, suggesting creative activities, and offering educational content tailored to the user’s interests. Meaningful occupation not only counters loneliness but also contributes to cognitive and emotional well-being.

Accessibility and Usability: In line with other authors [14,22], since older adults may have varying levels of technological literacy, the chatbot must be easy to use and understand. Measures such as larger fonts, clear language, and intuitive navigation are essential to ensuring that the technology is accessible to everyone, regardless of their familiarity with technological devices.

Security and Privacy: As pointed out in various studies [13,21], given older adults’ concerns about the security and privacy of personal information, the chatbot must incorporate robust security measures. This includes clearly explaining how user information will be handled, ensuring confidentiality, and protecting against potential cyber threats.

Incorporation of Practical Features, according to Lappromrattana & Sooraksa [15]. In addition to addressing the emotional dimension, the ideal chatbot should offer practical features. This includes reminders and alerts for daily needs, facilitating communication with family members, and providing useful information. The ability to adapt to individual preferences enhances the practical utility of the chatbot.

Active User Participation in Development has been brought to the forefront in numerous studies [19,22]. Actively involving older adults in the chatbot’s design and testing process is essential. This ensures that the chatbot effectively aligns with their needs and preferences, allowing for a more meaningful and satisfying user experience.

Encouragement of Social and Community Interactions, in line with Baker et al. [25]. The chatbot should encourage and facilitate participation in social interactions and virtual communities. This goes beyond one-on-one interaction with the chatbot, extending to the creation of broader connections and involvement in online social activities, thus countering loneliness by building a virtual support network.

### 5.2. Practical Recommendations for Interface Development

The following are a series of recommendations, based on our findings, focused on creating friendly, accessible, and empathetic interfaces, ensuring a positive and effective experience for older adults when interacting with chatbots.

Creating friendly interfaces to encourage active engagement of older adults with chatbots requires careful attention to various aspects, as emphasized in a variety of studies [18,23,24]. First, personalization emerges as a crucial element. Designers must ensure that chatbots can adapt to users’ personalities, interests, and cultural contexts. Offering customization options, such as conversation tone adjustments and content preferences, contributes to a more engaging and meaningful experience.

Usability and accessibility are fundamental considerations, in line with the study by Razavi et al. [14]. Interfaces should be designed with readability, clear language, and intuitive navigation in mind. It is essential to anticipate and address possible cognitive or physical limitations in older adults to ensure a frustration-free experience. Providing accessibility options, such as adjustable text sizes and voice command support, further enhances accessibility.

Clear Instructions and Tailored Feedback, as stated by several authors [19,22], are needed to overcome potential barriers related to anxiety and frustration. Feedback should be adapted to the user’s needs and preferences, ensuring a positive experience and minimizing the perception of difficulties in understanding instructions. On the other hand, natural and conversational interaction is needed [27]. The chatbot should be designed to offer a natural and conversational interaction, effectively mimicking human conversations. The ability to maintain fluid dialogues, understand the context of the conversation, and respond coherently contributes to establishing an emotional connection with users, making the interaction more valuable.

Adapted feedback is another key to successful interface design, according to several studies [14,19,22]. Incorporating feedback that aligns with the needs and preferences of the user, along with clear and emotionally intelligent conversational prompts, enhances the overall experience. User testing should be inclusive, with active participation from older adults in the design and testing process. Obtaining continuous feedback and making adjustments based on user experiences and suggestions ensures the relevance and effectiveness of the interface.

In terms of ease of use, it is imperative that chatbots are designed to be intuitive and understandable, especially for older adults with varying levels of technological literacy [23]. Clear and step-by-step instructions for chatbot functions contribute to a hassle-free experience. Additionally, content should be meaningful and engaging, with personalized recommendations and reminders aligning with users’ individual preferences [27].

Training and education on chatbot use, especially for those less familiar with technology, are essential [23]. Our findings reveal that providing educational resources facilitates adoption and effective use. Finally, establishing a continuous iteration process based on user feedback and evolutions in the needs of older adults ensures that interfaces remain relevant and effective over time. Together, these considerations are essential for creating interfaces that enhance the quality of life and emotional well-being of older adults.

### 5.3. Ethical Considerations in Chatbot Development

Ethical consideration in the design and application of chatbots for older adults is essential and is derived from various aspects highlighted in the reviewed studies. Ethical considerations in the development and application of chatbots for older adults revolve around respect, transparency, privacy, and equity, ensuring that these technologies enhance users’ lives in an ethical and responsible manner.

Ensuring informed and transparent consent is crucial. Older adults should receive clear information about how their data will be used, the nature of the interaction with the chatbot, and any possible impact on their privacy. Designers should strive to obtain informed consent before users engage in meaningful interactions.

The privacy and security of older adults’ personal information are fundamental ethical considerations [13,24]. Designers should implement robust security measures and ensure the confidentiality of data. Transparency in privacy policies and users’ ability to understand and control access to their data are crucial aspects.

Consideration of cultural and linguistic diversity is ethical. Chatbots should be sensitive to the cultural and linguistic differences of users, ensuring that interactions are inclusive and understandable for people from diverse backgrounds [12,20].

Accessibility is an ethical concern to ensure that chatbots are usable by people with various physical and cognitive abilities [14]. Designers should incorporate features that facilitate access for those with limitations, such as adjustable text options, voice commands, and an easy-to-use interface.

The inclusion of empathy and emotional sensitivity in interactions is ethical. Chatbots should be designed to understand and respond empathetically to users’ emotions, providing emotional support without crossing ethical or emotional boundaries [15].

Active user participation in the design and testing process is ethical [19,22]. Older adults should be involved meaningfully, providing their opinions and experiences to inform and improve chatbot development. This ensures that technological solutions adapt to their needs and preferences ethically.

Ensuring equity in access to technology is an ethical consideration. Designers should address the digital divide and ensure that chatbots are accessible to everyone, avoiding the exclusion of those who may be less familiar with technology [21].

Transparency and honesty in interaction are ethical principles. Users should clearly understand that they are interacting with a chatbot and not a human. Transparency about the capabilities and limitations of the chatbot is essential for building an ethical relationship with users.

### 5.4. Limitations

While our study endeavors to shed light on the potential of chatbots in alleviating loneliness among older adults, it is imperative to acknowledge certain limitations inherent in our research design and focus. Recognizing these constraints is crucial for a nuanced interpretation of our findings and for guiding future investigations in this evolving field. Our study, while informative, is not without its challenges, and a thoughtful consideration of these limitations enhances the transparency and comprehensiveness of our exploration.

The scope of our literature review, though thorough within its defined parameters, bears the imprint of limitations. The exclusion of publications in languages other than English and Spanish, along with the omission of gray literature, introduces a potential source of bias and restricts the holistic landscape of existing knowledge in this domain.

Our carefully curated sample, although diverse, prompts reflection on its representativeness across the broader spectrum of older adults, cautioning against wholesale generalizations. Accessing older adults in care facilities introduces logistical challenges, impacting diversity and sample representativeness, given the complexities arising from cognitive and physical decline. Cultural considerations permeate our findings, emphasizing the influence of cultural variations on chatbot acceptance. Insights gleaned are context-bound, urging circumspection when extrapolating results to diverse cultural milieus. 

This study deliberately focused on older adults living independently or in nursing homes to delve into their experiences with loneliness and technology, aiming for specific insights from this demographic. However, a notable limitation is the exclusion of perspectives from care home employees and relatives. Their input could have offered valuable context and a more comprehensive view of the challenges and solutions related to loneliness among older adults. Future research should consider including relatives to enrich this study with diverse viewpoints and provide a holistic understanding of the interplay between technology, loneliness, and the social environments of older adults.

Exploring the impact on specific subgroups within our diverse cohort prompts contemplation of tailored investigations. Deeper insights into variations across subgroups, spanning degrees of loneliness, cognitive abilities, or health conditions, could enrich the understanding of chatbot efficacy.

The temporal confines of our study underline the dynamic nature of external factors influencing technology. The rapid evolution of technology or societal attitudes could reshape the relevance and effectiveness of chatbot interventions, a consideration that shapes the trajectory of future research.

Our study predominantly focuses on short-term outcomes, leaving the long-term implications of chatbot interventions underexplored. The need for sustained investigations into the enduring effects on the well-being of older adults necessitates a deliberate extension of our temporal focus.

Acknowledging these limitations not only shapes the narrative of our present study but also lays a foundation for the nuanced evolution of future research endeavors, fostering a deeper understanding of chatbot interventions’ implications for addressing loneliness among older adults.

## 6. Conclusions

Our thorough research has provided a comprehensive understanding of the intersection between loneliness, emotional and social needs, and technology as a support tool for older adults. In this study, we have explored key trends in chatbot design, users’ emotional and social perceptions, technology adoption, and barriers that may impact its usage.

One highlighted element is the significance of personalization in chatbot design, which involves tailoring these tools to users’ personalities, interests, and cultural contexts. Furthermore, we have emphasized the essential active involvement of older adults in testing and developing these systems, ensuring that chatbots effectively meet their needs and preferences, contributing significantly to an improved quality of life and emotional well-being.

Users’ emotional and social perceptions emerge as fundamental factors for the successful acceptance of chatbots as companions. The ability of these systems to provide emotional support, companionship, and establish meaningful connections is crucial for their success and real utility in the lives of older adults.

In the realm of technology adoption, we have observed a general receptiveness toward using intelligent assistants for health information management. However, significant barriers such as affordability, technical issues, and concerns about privacy and security have been highlighted, which could hinder widespread adoption of these technologies among older adults.

In conclusion, our findings suggest that interacting with chatbots has the potential to play a positive role in mitigating loneliness in older adults. However, successful implementation must proactively address perceived barriers and consider the diversity of experiences and needs within this demographic. These results will not only inform future research but also guide the development of technology-based interventions focused on the emotional well-being of older adults.

Looking ahead, we urge deeper research to address identified gaps, such as the lack of standardization in intervention design and the need to explore the long-term consequences of older adults’ interaction with chatbots. Additionally, we emphasize the importance of paying closer attention to the social and emotional aspects of technology acceptance by older adults, advocating for an approach focused on designing solutions that adapt to their individual needs and preferences. These efforts will contribute to significant advancements in the field and, more importantly, tangible improvement in the quality of life for the older population.

## Figures and Tables

**Figure 1 healthcare-12-00062-f001:**
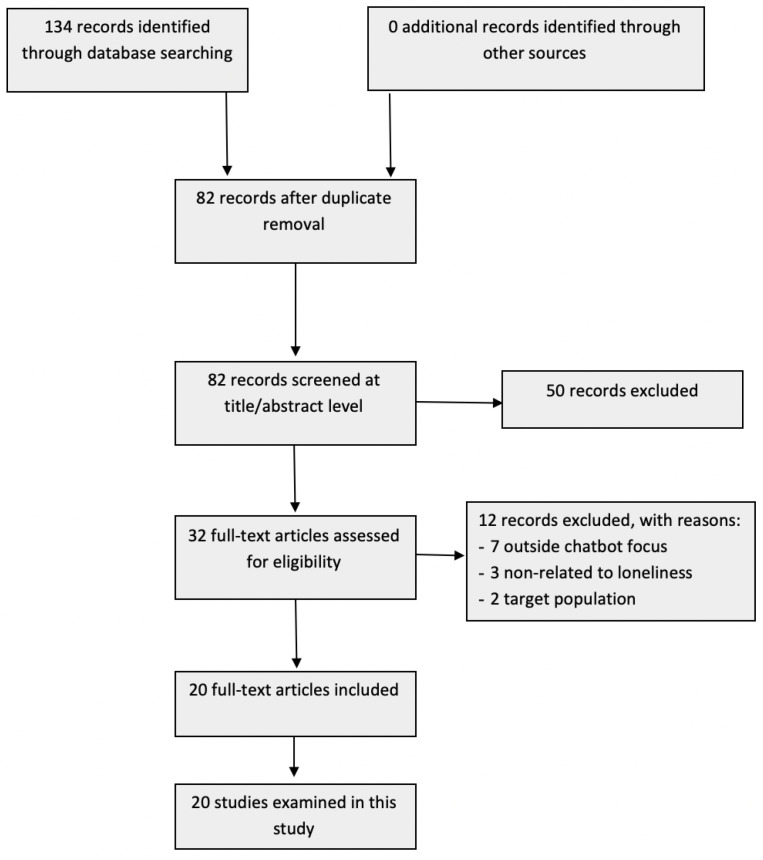
Reporting item flow chart (according to PRISMA, 2020 [11]).

**Table 1 healthcare-12-00062-t001:** Participant Characteristics: Demographic and Socioeconomic Profiles.

Characteristic	Nursing Homes	Private Homes	Focus Group 1	Focus Group 2
	(*n* = 20)	(*n* = 6)	(*n* = 8)	(*n* = 8)
Age Range				
65 to 74 years	30%	33.3%	50%	37.5%
75 to 80 years	70%	66.7%	50%	62.5%
Gender Distribution				
Male	40%	50%	50%	50%
Female	60%	50%	50%	50%
Marital status				
Widowed	80%	100%	62.5%	75%
Married	5%		12.5%	25%
Single	15%		25%	
Housing Arrangements				
Nursing Homes	100%			
Rented Apartments		50%	25%	62.5%
Owned Apartments		50%	37.5%	37.5%
Owned Single-family Houses			37.5%	
Education Levels				
Illiterate	20%			
Basic Literacy without Education	40%			25%
Primary Education	20%	33.3%		37.5%
High School or Equivalent	10%	33.3%	37.5%	37.5%
Higher Education/University	10%	33.3%	62.5%	
Work Experience				
Professional Roles	20%	100%	100%	
Itinerant Jobs	65%			100%
No paid Professions	15%			
Family Proximity	100%	83.30%	100%	87.50%

**Table 2 healthcare-12-00062-t002:** Trends, approaches, and gaps.

Topic	Description
Growing demand for solutions for older adults	There is an increasing demand for solutions to support the health, social participation, and independent living of the older adults’ population [12].
Technical focus in chatbot development	Research has focused on technical aspects such as natural language processing and machine learning algorithms [13].
Functionalities of chatbots in previous studies	Focus on functionalities such as reminders and medication management [13].
Current trends	There is an increase in the use of chatbots for health coaching and social support and an interest in personalization through natural language processing and machine learning [14].
Exploration of linguistic and emotional aspects	Use of linguistic and emotional features, as well as avatars and robots as companions for older adults [14].
Trend towards technological interventions for social isolation	Growing trend in using technology to address social isolation and loneliness in older adults [16].
Gaps in the representation of older adults in studies	Underrepresentation of older adults, especially those aged 85 and above, in research on the acceptance of social robots [18].
There is a need for research on attitudes towards social robots	Lack of a complete understanding of the attitudes and experiences of older adults towards social robots suggests the need for more research [18].
Focus on chatbot personalization	Personalization in chatbots for interaction with older adults involves adapting language and features to the user’s personality and needs [19].
Barriers to technology adoption	Barriers to technology adoption by older adults include difficulties with instructions and concerns about privacy and security [21].
Lack of standardization in technological interventions	There is a lack of standardization in the design and implementation of technological interventions, making it challenging to compare results across studies [16].
Need for research on long-term effects	There is limited research on the long-term effects of technological interventions and their effectiveness in different populations of older adults [16].
Lack of standardization in chatbot evaluation	There is a need for standardization in chatbot evaluation, requiring holistic analyses of their impact on the lives of older adults [19].
Research on chatbot accessibility	Research on chatbot accessibility for those with physical and cognitive limitations [19].
There is a need for research on emotional impact	There is a need for research on the emotional impact of chatbots on older adults, seeking solutions that provide meaningful and effective support [19].
There is a need for research on social and emotional aspects	There is a need for research on social and emotional aspects of designing technologies that adapt to the individual needs and preferences of older users [12].
Use of sentiment analysis	Use of sentiment analysis to assess the social acceptance of virtual assistants [12].
Need to explore ethical implications	We need to explore ethical implications in the use of virtual assistants, focusing on issues such as privacy, security, and autonomy [12].
Exploration of user behavior	There is a need for broader studies to better understand user behavior in chatbot environments, investigate the impact on social relationships, and explore ethical implications for older adults [14].

**Table 3 healthcare-12-00062-t003:** Design aspects.

Aspect of Design	Description
Robot Companion Functions	- Express emotions. - Understand emotions. - Initiate conversations. - Accompany in walks.
Emotional and Social Support in Chatbots	- Incorporate features that provide emotional support. - Encourage social interaction.
Frustration and Conversation Limitations	- Consider the chatbot’s capacity for real conversations. - Avoid restrictions in social communications.
Attractiveness and Empathy in Design	- Design chatbots to be attractive and supportive. - Incorporate emotional intelligence.
Conversational Stimuli and Feedback	- Use conversational stimuli tailored to your needs and preferences. - Provide tailored feedback.
Accessibility and Usability	- Implement accessibility measures (larger fonts, clear language, intuitive navigation). - Facilitate use for older adults with varying tech literacy levels.
Personalization of Interaction	- Adapt to personality, interests, and cultural context. - Provide personalized recommendations and reminders. - Adjust language and style based on individual preferences.
Intuitive and Accessible Design	- Implement simple and clear interfaces. - Provide concise instructions. - Use technology to enhance accessibility, especially for those with physical or cognitive limitations.
Relevance of Emotional Support	- Offer companionship and emotional support. - Particularly valuable for those experiencing social isolation or loneliness.
Active Involvement of Older Users	- Engage older adults in the design and testing process. - Ensure chatbots meet their needs and preferences.
Feedback and User Testing	- Prioritize user feedback and testing. - Ensure chatbots are effective and meet user needs.
A Comprehensive Approach to Development	- Key strategy for creating engaging and meaningful experiences. - Contributes to the successful development of chatbots, enhancing quality of life and emotional well-being.

**Table 4 healthcare-12-00062-t004:** Summary of perceptions, technology adoption, and barriers.

**Emotional and Social Perceptions**
Acceptance is positively influenced by exposure and limited interaction with social robots. Attribution of emotional and social functions to companion robots. User behavior analysis related to avatars includes expressiveness, emotions, and personal revelation. Adaptation of chatbots to meet the needs and preferences of older adults. Influence of attitudes toward technology on older adults, emphasizing utility, ease of use, and social acceptance. Technology is perceived as a facilitator for sharing health information and improving communication. There are varied user perceptions of relationships with chat agents, from superficial to a sense of companionship. Older users’ potential anxiety and frustration with chatbots is linked to technology unfamiliarity and privacy concerns.
**Technology Adoption and Utility**
Positive acceptance of companion robot functions among older adults, regardless of previous technological experience. Older adults’ general receptivity to using intelligent assistants for health information management. There is a desire for personalized, intelligent assistants in health management among older adults. Positive emotional response to the idea of using technology to improve health outcomes. There are concerns about the accuracy and reliability of information provided by intelligent assistants for health management. Recognition of chatbots’ potential to provide emotional support and companionship to address social isolation. Chatbots’ role in fostering social interaction and engagement, countering isolation and loneliness. Chatbots’ contribution to cognitive health is through the provision of tailored cognitive exercises and educational content based on user interests.
**Barriers**
Concerns about affordability, potential technical difficulties, privacy, and security are barriers to the adoption of social robots among older adults. Perceived barriers to technology adoption, including privacy and security concerns and a lack of familiarity with technology. Desire for easy-to-use and understandable intelligent assistants, indicating a perception of technology as complex or difficult to navigate. Perceived barriers to technology adoption among older adults, including the stigma associated with technology, threats to autonomy and privacy, and insufficient information and training. Chatbots are potential solutions to overcome barriers to technology adoption among older adults by providing intuitive interfaces, clear instructions, and personalized recommendations.

**Table 5 healthcare-12-00062-t005:** Identified the needs of older adults.

Need	Details and Examples
Company and Social Connection	- Company and social connection: The importance of meaningful interactions with family and friends. - Cultivating new relationships: Expanding social circles to counteract the loss of connections. - Participation in social activities: Clubs, local events to foster connection.
Emotional Support	- Emotional support: The need for emotional support in difficult times and in daily life. - Sharing with family and friends: The importance of expressing thoughts and emotions with trusted individuals. - Emotional support through technology: Interest in technological solutions to offer sensitive and empathetic companionship.
Meaningful Occupation	- Meaningful occupation: Engaging in meaningful and rewarding activities to combat loneliness. - Challenge, stimulation, and achievement: Activities that provide a sense of purpose and contribute to emotional well-being. - Participation in virtual communities: Through technology, connect with others with similar interests.
Access to Health and Well-Being Resources	- Access to health resources: Mobility and proximity to medical services as key variables. - Technology to overcome barriers: Online platforms, virtual consultations for those with mobility limitations. - Up-to-date health information: Desire for resources providing practical knowledge about healthy aging.
Safe Environment and Personal Care	- Safety in the environment: The importance of living in safe and comfortable places. - Personal care and caregiver companionship: Support in daily activities and companionship are essential for emotional well-being. - Technology for security: Valuation of technological solutions for quick access to help and monitoring devices.
Technological Adaptability	- Attitudes toward technology: A variety of attitudes, from positive openness to resistance or anxiety. - Accessibility and usability: Importance of simple and user-friendly interfaces. - Training and support in technology use: Guidance to improve confidence and willingness. - Perception of benefits: Recognition of advantages such as staying connected and participating in online activities.

## Data Availability

The qualitative dataset and transcriptions of narratives are not publicly available due to ethical restrictions and privacy issues.

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
