# Peer review of "Qualitative Analysis of Conversational Chatbots to Alleviate Loneliness in Older Adults as a Strategy for Emotional Health"

_healthcare, 2023, doi:10.3390/healthcare12010062_

Round 1

Reviewer 1 Report

Comments and Suggestions for Authors

The authors have chosen to holistically address the complex issues surrounding the use of chatbots as companions for older adults and to help them overcome their loneliness. They stated three purposes of their investigation. The prime purpose is ‘a detailed and enriching exploration of conversational chatbots specifically designed to address the issue of loneliness of older adult population’ (lines 79-80). Second purpose is emphasizing ‘how these systems can transcend the technological realm to become understanding and empathetic allies that substantially contribute to the emotional well-being and socialization of this population’ (lines 87-89) and ‘positive impact on overall health and quality of life’. Authors decided to explore the ethical issues as protection of privacy and autonomy of older adults.

The research methodology is correct and includes a research phase (Review of Previous Approaches - line 124) and the qualitative studies (interviews and focus groups with older adults). Description of the research results concludes the evaluation of the chatbots used by the older adults.

Areas of strength:

Innovative approach to supporting older adults through AI-based technologies

Clearly stated purposes of the article

Correct terminology for 'older population'; the authors decided to use the term 'older adults' which is in line with the promoted correctness of expression

Areas of weakness:

Introduction – complete the introduction with one sentence on the purpose of the article. There are objectives stated in part No 2 of the article but try to consider to slot the purpose of the article

Inadequate description of the research trial; the only information is – the sample consists of 26 older adults of both genders, aged between 65 and 80 selected in a stratified manner to represent diverse socio-economic and cultural experiences (line 186-188). That means we know nothing about the investigated people. Do they live in care homes, are they single or not etc. – these factors [and others] influence their answers.

Focus groups – each group consisted of eight participants, older adults of both genders, aged between 65 and 80) – THAT IS ALL. We may suppose that the focus groups consist of the same people who were investigated; However, other people with experience in dealing with loneliness among older people, such as employees of carehomes/ homes for the elderly, were not invited to the focus groups. Any reflection –why?

Author Response

We sincerely appreciate your thoughtful and constructive feedback on our manuscript. Your insightful suggestions have significantly enhanced the clarity and rigor of our study. We are grateful for the time and effort you dedicated to reviewing our work.

We have carefully considered each of your comments and have implemented the recommended changes. Your guidance has been invaluable in refining the focus, organization, and language of our manuscript. We believe these revisions have strengthened the overall quality and contribution of our study.

To facilitate your review of the changes, all modifications in the manuscript are highlighted in blue.

Point 1.

The authors have chosen to holistically address the complex issues surrounding the use of chatbots as companions for older adults and to help them overcome their loneliness. They stated three purposes of their investigation. The prime purpose is ‘a detailed and enriching exploration of conversational chatbots specifically designed to address the issue of loneliness of older adult population’ (lines 79-80). Second purpose is emphasizing ‘how these systems can transcend the technological realm to become understanding and empathetic allies that substantially contribute to the emotional well-being and socialization of this population’ (lines 87-89) and ‘positive impact on overall health and quality of life’. Authors decided to explore the ethical issues as protection of privacy and autonomy of older adults.

The research methodology is correct and includes a research phase (Review of Previous Approaches - line 124) and the qualitative studies (interviews and focus groups with older adults). Description of the research results concludes the evaluation of the chatbots used by the older adults.

Areas of strength:

  • Innovative approach to supporting older adults through AI-based technologies
  • Clearly stated purposes of the article
  • Correct terminology for 'older population'; the authors decided to use the term 'older adults' which is in line with the promoted correctness of expression

Response to point 1:

Thank you for your positive feedback and insightful observations. We appreciate your acknowledgment of the innovative approach we have taken to support older adults through AI-based technologies. Your recognition of the clearly stated purposes of our article is encouraging. We have carefully considered your comments and will continue to refine our work based on your valuable input.

Point 2.

Areas of weakness:

  • Introduction – complete the introduction with one sentence on the purpose of the article. There are objectives stated in part No 2 of the article but try to consider to slot the purpose of the article

Response to point 2:

Thank you for your thoughtful feedback. We have carefully considered your suggestion and have enhanced the introduction by incorporating a concise sentence that explicitly outlines the purpose of the article. This addition aims to provide readers with a clear and succinct overview of the article's overarching objective. We appreciate your guidance and are committed to continuously improving the clarity and structure of our work.

Point 3.

  • Inadequate description of the research trial; the only information is – the sample consists of 26 older adults of both genders, aged between 65 and 80 selected in a stratified manner to represent diverse socio-economic and cultural experiences (line 186-188). That means we know nothing about the investigated people. Do they live in care homes, are they single or not etc. – these factors [and others] influence their answers.

Response to point 3:

Thank you for your insightful comments. We have taken into consideration your valuable feedback regarding the description of the research trial. In response, we have enhanced the description by providing additional details about the participants, including factors such as their living arrangements. This information aims to offer a more comprehensive understanding of the context in which the study was conducted and the potential influences on participants' responses.

Point 4.

  • Focus groups – each group consisted of eight participants, older adults of both genders, aged between 65 and 80) – THAT IS ALL. We may suppose that the focus groups consist of the same people who were investigated; However, other people with experience in dealing with loneliness among older people, such as employees of care homes/ homes for the elderly, were not invited to the focus groups. Any reflection – why?

Response to point 4:

Thank you for your perceptive observations. In response to your feedback, we have expanded the description of the focus groups, providing additional details on the composition of each group.  It is essential to clarify that the focus groups and interviews involve distinct participant groups, as explicitly outlined in the methodology section.

The composition of the study sample intentionally focused on older adults living independently or in nursing homes to gain insights into their experiences with loneliness and technology. This choice was driven by the aim to explore the specific needs and perspectives of this demographic in addressing loneliness.

While this approach allowed for a deep understanding of the targeted group, it's important to acknowledge the limitation of not including perspectives from employees in care homes and relatives. Their insights could have provided valuable context and a more comprehensive view of the challenges and potential solutions related to loneliness among older adults. Employees in care homes interact closely with residents and could offer a unique perspective on the effectiveness of technology interventions and other strategies.

In future research, the inclusion of relatives would enhance the study's richness by incorporating the views of those who directly support older adults in a different living context. This would contribute to a more holistic understanding of the complex interplay between technology, loneliness, and the varied social environments of older adults.

Reviewer 2 Report

Comments and Suggestions for Authors

Please, see the attached review.

Comments on the Quality of English Language

English language is good. Minor adjustments are required.

Author Response

Response to Comments of Reviewer 2

We sincerely appreciate your thoughtful and constructive feedback on our manuscript. Your insightful suggestions have significantly enhanced the clarity and rigor of our study. We are grateful for the time and effort you dedicated to reviewing our work.

We have carefully considered each of your comments and have implemented the recommended changes. Your guidance has been invaluable in refining the focus, organization, and language of our manuscript. We believe these revisions have strengthened the overall quality and contribution of our study.

To facilitate your review of the changes, all modifications in the manuscript are highlighted in blue.

Point 1.

TITLE. Analysis of Conversational Chatbots to Alleviate Loneliness in 2 Older Adults as a Strategy for Emotional Health

This study explores the topic of conversational chatbots designed as innovative solutions to alleviate loneliness in older adults, thus positively contributing to their health and quality of life, emotional well- being and socialization. A mixed methodology is used (literature review and qualitative research). Issues regarding privacy, autonomy, usability, and technology acceptance are addressed. The findings suggest that interacting with chatbots has the potential to play a positive impact on older adults, even though a successful implementation should consider related barriers and the cultural diversity of users. These results could guide the development of technology interventions focusing on the emotional support to older adults.

General comments. The paper is well structured, developed, and detailed. However, some terms/concepts seem sometimes redundant, and moreover some integrations/clarifications/definitions seem needed, especially in Materials-Methods and Results sections.

Response to point 1:

Thank you for your feedback and suggestions. Your comments will help us improve the article, particularly by clarifying terms, methodology, and avoiding redundancy.

Point 2.

Title: Something on the method could be included (e.g. literature review and qualitative analysis)

Keywords. Authors could add “qualitative study”

Response to point 2:

Thank you for your thoughtful consideration of the title. We agree that the title should convey information about the methodology used. Given that our method incorporates various types of methodologies, we find it appropriate to include the term "Qualitative Analysis." Additionally, we have included the keyword "Qualitative Study" to further emphasize the nature of our research.

Point 3.

Introduction. This section is well done. I would however suggest the following.

  • line 30: to define loneliness (mainly subjective perception, feeling of being alone/neglected) and add a reference in this respect;
  • lines 33 and 37-38; lines 75-77: these sentences (To address this issue, the use of chatbots has been explored; Throughout this article, we will explore the development of conversational chatbots as a solution to alleviate loneliness in older adults; This is the primary motivation for conducting this study: to gather information that will help conversational chatbots become allies for older adults, somehow mitigating their loneliness) relate more to the scope of the paper. Thus they could be moved to par. 2. Objectives;
  • line 47: to define social isolation (mainly objective context, few contacts with others) and add a reference in this respect.

Response to point 3:

Thank you for your insightful comments. We have incorporated the definitions of "loneliness" and "social isolation," along with their respective references. These sentences have been integrated at the beginning of section 2. Objectives, excluding them from section 1. Introduction. Additionally, following the suggestion of another reviewer, we have added a paragraph at the end of section 1, outlining the purpose of the article. This paragraph serves as a bridge to the specified objectives, which are then detailed afterward.

Point 4.

Materials and Methods. This section is very detailed, but some parts are a bit redundant. I would suggest to reduce it by avoiding particularly repeated concepts. E.g. the aim of the review is described at line 115, at lines 125-126, and in par. 3.1. The same criticality regards the description of the qualitative research and chatbot evaluation. In other words, I would suggest to integrate the first part of this section (lines 107-149) in the paragraphs below (3.1…..3.3) since they present/repeat the same concepts.

Response to point 4:

Thank you for your valuable feedback. We have addressed your suggestion by reducing the introduction in the Materials and Methods section. The relevant information has been integrated into the appropriate subsections, streamlining the content and minimizing redundancy. Specifically, we have retained only the information related to the methodology from a global perspective. We believe this adjustment enhances the clarity and conciseness of the methodology section.

Point 5.

Moreover, several/mixed methods are used (literature review, qualitative research/focus groups and interviews; chatbot evaluation), but something seems missing in their description. For instance:

  • I cannot see the PRISMA template for the literature review, whose details are only mentioned in the text of the paper, par. 4.1, lines 254-260;
  • I cannot see a scheme/table including relevant data that were extracted from each included study (author, publication year, study design, and so on). These are only mentioned in the text of the paper, par. 3.1.

Response to point 5:

Thank you for your valuable comments. We appreciate your suggestions on the nature of literature reviews, and we concur with the distinction made by Grant and Booth (2009): A literature review is a generic term that may or may not include exhaustive searches; the synthesis is typically narrative, and the analysis can be conceptual. As indicated, our intent with this literature review is to provide a comprehensive overview and updated insights into the use of conversational chatbots with the older adult population, particularly in alleviating loneliness. In any case, conducting a narrative synthesis on trends, prior approaches, and identified gaps, not necessarily from all the works obtained in the search, but from studies relevant to the broader context of the research.

In response to your suggestion, we will include the PRISMA diagram to enhance the transparency and rigor of our methodology. However, we think that a detailed table with author, year of publication, design, etc., may not be necessary for this study. Instead, we have opted for a synthesis table focusing on trends, approaches, and identified gaps in relevant studies. We are open to including a more detailed table if deemed appropriate by the reviewer.

Finally, we have diligently improved the methodological description by incorporating pertinent details, also addressing suggestions from another reviewer.

Point 6.

Further comments:

  • Authors state that search strategy for literature was meticulously designed and systematic (par. 3.1), but data were analysed through a narrative and thematic approach. I think they should define better the type of review (systematic/scoping-narrative review?);
  • line 119: a representative sample of older adults was utilized. I would suggest to explain better why it was considered representative (was it purposive? For instance, in a purposive sampling units/cases are selected due to particular characteristics allowing a deep exploration of the phenomenon of interest);
  • line 124: the type of review should be indicated (systematic/scoping-narrative);
  • lines 206 and 990: the type of informed consent (written or verbal) should be specified;
  • 3.2: Authors could explain how they combined/analysed results from interviews and focus groups. Looking at Appendix A. (Semi-Structured Interview Design), and Appendix B. (Focus Group Design), some explored topics are the same.

Response to point 6:

We express our sincere gratitude for the comments provided by the reviewer, which have significantly contributed to refining and enhancing the quality of our manuscript. All the suggestions have been incorporated to the manuscript.

As mentioned earlier, a literature review has been conducted with the stated objective. While systematic review steps have been followed to enhance rigor, the purpose is to gather supporting information through a narrative and thematic approach. Therefore, the approach used is that of a narrative review (literature review with a narrative and thematic approach).

The nature of the sample was designed to ensure representation, encompassing various characteristics of both interviewees and focus groups. Now this nature of the sample is described in the manuscript.

The type of review is now defined and justified more effectively in Section 3.1.

Informed consents were in written form but assisted by the interviewer, who read and explained them to individuals with greater difficulty, consistently checking for their comprehension. In addition to obtaining verbal consent and ensuring clarity, participants were then asked to sign the consent form. The type of informed consent is now added to the manuscript.

The results obtained from interviews and focus groups were processed using ATLAS.ti 23 in a uniform manner. The findings from both methods were integrated, employing consistent classification criteria for categories since the questions, while appropriately tailored to interviews or focus groups, shared similarities. Notably, the older adults selected for interviews and focus groups were distinct, meaning that the same sample was not used for both data collection methods. This aspect is now elucidated more clearly in the methodology section.

Point 7.

Results. The text of this section is almost detailed, but some concepts are repeated, with tables however well summarizing the findings of the literature review. I would firstly suggest to enrich the qualitative analysis by reporting (if possible/relevant) some quotations/verbatim extracts from narratives/interviews, to support the overall picture.

Response to point 7:

Thank you for your valuable feedback, which we greatly appreciate. In the same way as the reviewer's suggestion, we believe that incorporating some quotations or verbatim extracts from narratives and interviews can provide stronger support for our results. Consequently, we have included relevant excerpts in key sections of the results to enhance the robustness of our findings. Repeated concepts have been reviewed.

Point 8.

Moreover, I would suggest the following:

  • 4.1.1, line 268: sentiment analysis should be defined, and a related reference added;
  • par 4.2. Qualitative Research: to integrate the first part of this section (lines 472-514) in the paragraphs below (4.2.1…..4.2.6) since these parts present similar/same concepts;
  • 4.2, lines 480-485: these regard the method of qualitative analyses thus they should be moved to method (par. 3.2.). In addition, if ATLAS was used also to analyse interviews (besides focus groups) should be specified;
  • at par. 4.2.2 and 4.2.5 it is indicated that results here presented come from interviews, and at par. 4.2.4 and 4.2.6 it is specified that results here presented are based both on the analysis of focus groups and interviews. Nothing it is specified in this respect for results presented at par. 4.2.1 and 4.2.3. I think it could be better/clearer to adopt only a method for reporting this information, that is, to specify in each paragraph the source of the qualitative findings (interviews or/and focus groups), or conversely avoid this details overall in these paragraphs, in case Authors agree with the suggestion above (i.e., in par. 3.2 Authors could explain/anticipate how they combined/analysed results from interviews and focus groups);
  • 4.3. Chatbot Evaluation: Authors state that selected chatbots have been identified based on a review of existing literature. I do not understand if they refer to their literature review (presented in the paper) or to other literature in general;
  • par 4.3.1. CATe Chatbot: when Authors refer to the evaluation results of this tool (i.e., older adults generally enjoyed talking to CATe and liked the application, achieving an overall satisfaction score of 3.98 out of 5), I suppose they refer to results from other authors cited in the paragraph (Sidner et al., 2018; Bravo et al., 2020), and not to their own evaluation. Please, clarify, in order to differentiate own evaluation by Authors of this paper (who state “In this section, we will proceed to conduct a detailed evaluation of various conversational chatbots”) and the one provided by other authors;
  • it could be useful to provide (if any) a link also for CATe Chatbot and Charlie Chatbot (as for ElliQ Chatbot).

Response to point 8:

Thank you for your thoughtful comments and suggestions. We appreciate your insights on the following aspects:

Sentiment analysis definition: The sentiment analysis mentioned, focusing on positive and negative scores of sentences as defined by Thakur & Han (2018), is considered beyond the scope of our current work, and including such a definition is deemed unnecessary. However, for clarity and reference, we have added the relevant citation from Thakur & Han (2018) to support this statement.

Qualitative research integration: As recommended, we have seamlessly integrated the majority of the first part of the indicated section into the relevant sub-sections, enhancing the overall flow and coherence of the manuscript.

Method of qualitative analysis: Apologies for any confusion in our initial expression regarding the use of ATLAS.ti. We have now clearly specified the application of ATLAS.ti for analyzing both interviews and focus groups, providing a more accurate representation of our methodology.

Method for reporting information: Following your suggestion, we have included details in the methodology section on how information is integrated. In the results section, we have adopted an approach that avoids detailed information about the source, promoting clarity and readability.

Identification of chatbots: Apologies for any confusion. Two of the chatbots were identified through the literature review conducted in this study, while the third one was already known to the authors through alternative sources. We have now appropriately explained this in our revised manuscript.

In reference to the mentions of the CATe evaluation: Now, in the specified evaluation, acknowledgment is made to the study by Bravo et al. (2020). We have adjusted the paragraph to eliminate any potential confusion.

Links for CATe and Charlie chatbots: We have verified our information, and indeed, no links to commercial chatbots were found. We appreciate your diligence in checking this aspect.

Point 9.

Discussion. This section is well done. I would however suggest the following:

  • to distinguish better results from the study of the Authors, and those from other/previous literature. For instance, I would generally suggest to write “our study/findings” to indicate the former, and to add “from literature”, “according to other authors” (or something similar) to indicate the latter. This also with regard to Results section;
  • 5.1.: Authors could specify better they present the possible features of an ideally designed chatbot based on the overall study's findings (literature review, qualitative research, and chatbot evaluation);
  • 5.1.: in some lines (e.g., 811, 813, 818, 825, 830, 835, 839), the colon could be replaced by the full stop;
  • maybe it could be better to integrate par. 5.1 (Ideal Chatbot Based on Conducted Studies), par. 5.2. (Practical Recommendations for Interface Development), and par. 5.3 (Ethical Considerations in Chatbot Development), since several aspects are repeated (e.g., accessibility and usability, personalization, adaptability, anxiety, meaningful content, privacy,…). In this respect I would suggest to provide only a new par. 5.1. as overall recommendations to develop an ideal and ethical chatbot.

Response to point 9:

Thank you very much for your valuable suggestions regarding distinguishing our work from others. We believe these changes will contribute to a clearer distinction between our contributions and those based on the work of other authors. The manuscript has been revised accordingly, and missing references have been incorporated.

The characteristics of the ideal chatbot design based on the findings of the study are now presented more effectively in section 5.1. We appreciate your feedback, and the use of colons has been replaced with full stops to maintain consistency with the rest of the paragraphs in various subsections.

We acknowledge your comment regarding the integration of subsections 5.1, 5.2, and 5.3. While we initially considered integrating them, given that certain aspects were repetitive across sections, we ultimately decided to keep them separate. Each section has its own distinct focus, especially in the case of recommendations for interface design and ethical considerations. Although some aspects are cross-cutting, appearing in different sections, we believe maintaining separate sections provides clarity. Thank you for your understanding.

Point 10.

Limitations. I would encourage Authors to think about possible limitations of their work, for instance, the search for publications to be included in the literature review could not be exhaustive (due to the exclusion of grey literature, of papers published in other languages than English and Spanish, due to the use of a limited number of databases for the search…).

Point 11.

Conclusions. This section is consistent with the evidence presented.

Institutional Review Board Statement. Authors state that ethical review and approval were waived for this study due to the fundamental principle of non-intrusive observation. However, they could at least add that the study was conducted in accordance with the Helsinki Declaration.

Data Availability Statement. Authors state “No underlying data were collected or produced in this study”. But they refer to interviews and focus groups, thus narratives were transcribed and analysed (these are qualitative data). Authors could, for instance, state that the qualitative dataset/transcriptions of narratives are not publicly available due to ethical restrictions and privacy issues (if this is the case). Or other sentences can be provided, corresponding to the real situation.

Response to point 11:

Thank you for your suggestions. These have been appropriately incorporated into the manuscript.

Point 12.

References are updated. However, these should be in the right format (numbers in square brackets in the text, and in the style requested by the Journal in the final reference list at the end of the paper).

English language is good.

Response to point 12:

Thank you for your feedback. We have modified the citations and references to align with the indicated format. We appreciate your comment on the quality of the English language; we have made efforts to ensure clarity and precision in our writing.

Round 2

Reviewer 2 Report

Comments and Suggestions for Authors

Authors did really a great and accurate work. Overall the suggested integrations and clarifications have been provided.

Comments on the Quality of English Language

Minor editing of English language is required.